



# Technical note: Novel estimates of the leaf relative uptake rate of carbonyl sulfide from optimality theory

Georg Wohlfahrt[1], Albin Hammerle[1], Felix M. Spielmann[1], Florian Kitz[1], Chuixiang Yi[2,3]

[1]Department of Ecology, University of Innsbruck, Innsbruck, 6020, Austria
5 [2]School of Earth and Environmental Sciences, Queens College, City University of New York, New York 11367, USA.
[3]Earth and Environmental Sciences Department, Graduate Center, City University of New York, New York, NY 10016, USA.

*Correspondence to*: Georg Wohlfahrt (georg.wohlfahrt@uibk.ac.at)

**Abstract.** In order to estimate the gross primary productivity (GPP) of terrestrial ecosystems from the canopy uptake of carbonyl sulfide (COS), the leaf relative uptake rate (LRU) of COS with respect to carbon dioxide needs to be known *a priori*. 10 Currently, the variability of the LRU between plant species in different biomes of the world is poorly understood, making the choice of an appropriate LRU uncertain and hampering further progress towards developing COS as an alternative tracer of GPP. Here we propose a novel approach for estimating LRU based on plant optimality principles, validate it against *in situ* leaf gas exchange measurements and provide global monthly climatological estimates. The global vegetation season average simulated LRUs fall into the range of 0.5-1.4 and are thus lower than any other published global estimates. We advocate these 15 LRU estimates to be adopted by global modellers in order to test to what degree these are compatible with our current understanding of the sources and sinks in the global COS budget.

## 1 Introduction

The gross primary productivity (GPP) is a key conceptual term in the ecosystem carbon cycle, however cannot be directly measured at ecosystem-scale, requiring the application of indirect approaches based on the combination of proxy 20 measurements and modelling (Wohlfahrt and Gu, 2015). During the last decade, carbonyl sulfide (COS) has emerged as a promising proxy for GPP, based on the observation that COS co-diffuses into plant leaves together with carbon dioxide ($CO_2$) during photosynthesis, but in contrast to the latter is not re-emitted (Sandoval-Soto et al., 2005).

The leaf relative uptake rate of COS with respect to $CO_2$, abbreviated as *LRU*, is instrumental to using COS as a proxy for GPP (Wohlfahrt et al., 2012). The *LRU* is the dimensionless ratio of the deposition velocities, that is the flux ($F$; pmol m$^{-2}$ s$^{-1}$ 25 and μmol m$^{-2}$ s$^{-1}$, respectively) normalized by the ambient (subscript *a*) mole fraction ($C$), of COS (superscript *s*; pmol mol$^{-1}$) with respect to $CO_2$ (superscript *c*; μmol mol$^{-1}$):

$$LRU = \frac{\frac{F^s}{C_a^s}}{\frac{F^c}{C_a^c}} .$$ (1)



Assuming negligible daytime mitochondrial leaf respiration (or accounting for it) allows replacing $F^c$ with GPP and, provided *LRU* is known, rearrangement of Eq. (1) then yields a framework for estimating GPP based on measurements of $C_a^c$, $C_a^s$ and

$F^s$ (Campbell et al., 2008):

$$GPP = F^s \frac{C_a^c}{C_a^s} LRU^{-1} \ . \tag{2}$$

Initial studies on the *LRU* suggested its value to gravitate to ca. 1.6 (Stimler et al., 2011; Stimler et al., 2010; Berkelhammer et al., 2014), a value which was successfully used by Asaf et al. (2013) in the first ever study that estimated ecosystem-scale GPP from corresponding COS flux measurements. The most recent review of published *LRU* values (Whelan et al., 2018)

however indicates that, even though the median *LRU* amounts to 1.7, 95 % of the values fall into the range of 0.7 – 6.2, which is consistent with theoretical back-of-the-envelope calculations by Wohlfahrt et al. (2012). Here it should be noted that some of the higher values may result from measurements under low, non-saturating, light conditions, which are known to cause LRU to increase (Kooijmans et al., 2019). More recently, two field studies (Kooijmans et al., 2019; Sun et al., 2018) reported values around 1 under high incident photosynthetically active radiation (PAR).


Replacing the flux terms in Eq. (1) with the underlying Fick's diffusion equations (see Seibt et al., 2010 for a derivation), yields Eq. (3), which allows an assessment of the drivers underlying the *LRU*:

$$LRU = 1.21^{-1} \left(1 + \frac{g_s^s}{g_i^s}\right)^{-1} \left(1 - \frac{C_i^c}{C_a^c}\right)^{-1} , \tag{3}$$

where $g_s^s$ and $g_i^s$ represent the stomatal and internal, respectively, conductances to COS (mol m$^{-2}$ s$^{-1}$), $C_i^c$ the CO$_2$ mole fraction

in the leaf intercellular space (µmol mol$^{-1}$) and the factor 1.21 converts the stomatal conductance to COS to its CO$_2$ counterpart. Note that the boundary layer conductances for COS and CO$_2$ have been assumed to be infinite here, as is typically the case in vigorously ventilated leaf chambers (Seibt et al., 2010). Eq. (3) shows the *LRU* to depend on two dimensionless ratios: (i) the stomatal-to-internal conductance for COS and (ii) the internal-to-ambient CO$_2$ mole fraction ratio. While the magnitude and drivers of $g_i^s$ are poorly understood, $g_s^s$ and $\frac{C_i^c}{C_a^c}$ are well known to vary over short timescales in response to diel changes in

environmental drivers, as well as along large-scale bioclimatic gradients (Lloyd and Farquhar, 1994). With regard to the former, recent work by Kohonen et al. (2022) and Sun et al. (2022) demonstrated that contrasting leaf gas exchange theories are able to reproduce and explain the observed short-term response of *LRU* to key drivers such as incident PAR or the vapor pressure deficit (VPD).

In contrast, variability in *LRU* between biomes is poorly understood, partially due to a scarcity of measurements (Whelan et al., 2018), partially due to the lack of a suitable theoretical framework, and the motivation for this work is thus to propose and apply a new theoretical approach for estimating large-scale bioclimatic patterns of *LRU*. To this end we make use of recent developments in plant optimality theory (Harrison et al., 2021).





## 2 Methods

### 2.1 Model

Here we use the P-model as described by Mengoli et al. (2022) and refer to this paper and references cited therein for further details. Briefly, the model, applicable only to C₃ plant species, is based on the combination of two optimality hypothesis – the least-cost and the coordination hypothesis. The least-cost hypothesis (Prentice et al., 2014) proposed that plants balance the carbon costs (per unit of photosynthesis) of maintaining the transpiration stream with those required for maintaining the carboxylation capacity and yields a $\frac{c_i^c}{c_a^c}$ ratio under which this balance is optimally realized:

$$\frac{c_i^c}{c_a^c} = \frac{\Gamma^*}{c_a^c} + \left(1 - \frac{\Gamma^*}{c_a^c}\right)\frac{\xi}{\xi + \sqrt{D}} \text{, with} \tag{4}$$

$$\xi = \sqrt{\frac{\beta(K_m + \Gamma^*)}{1.6\eta^*}}\,. \tag{5}$$

Here $\Gamma^*$ represents the CO₂ compensation point (Pa) in the absence of mitochondrial respiration, $D$ the VPD (Pa), $K_m$ the effective Michaelis-Menten coefficient of RUBISCO (Pa), $\eta^*$ the dimensionless ratio of the viscosity of water at a given temperature to that at 25°C and $\beta$ is a calibrated constant (146) representing the ratio of the two cost terms. $\xi$ (Pa$^{0.5}$) represents the VPD response of the $\frac{c_i^c}{c_a^c}$ ratio. Eq. (4) has been successfully validated against global $\frac{c_i^c}{c_a^c}$ ratios derived from C$^{13}$ isotope data by Wang et al. (2017b).

The coordination hypothesis (Maire et al., 2012) assumes that plants coordinate the investment of resources into electron transport and carboxylation capacity in a way such that photosynthesis, under average environmental conditions, is co-limited by the two and yields optimal values of the maximum carboxylation rate ($V_{Cmax}$; μmol m$^{-2}$ s$^{-1}$) and the maximum electron transport rate ($J_{max}$; μmol m$^{-2}$ s$^{-1}$):

$$V_{Cmax} = \varphi_o I \frac{c_i^c + K_m}{c_i^c + 2\Gamma^*} \sqrt{1 - \left[c^* \frac{c_i^c + 2\Gamma^*}{c_i^c - \Gamma^*}\right]^{2/3}} \text{, and} \tag{6}$$

$$J_{max} = \frac{4\varphi_o PAR}{\sqrt{\frac{1}{1 - \left[c^* \frac{c_i^c + 2\Gamma^*}{c_i^c - \Gamma^*}\right]^{2/3}} - 1}}\,. \tag{7}$$

Here $\varphi_o$ stands for the intrinsic quantum efficiency of photosynthesis (mol mol$^{-1}$), $I$ represents PAR (μmol m$^{-2}$ s$^{-1}$) and $c^*$ is a calibrated (0.41) dimensionless cost factor for electron transport. $V_{Cmax}$ was successfully validated against corresponding leaf gas exchange measurements by Smith et al. (2019).





$V_{Cmax}$ and $J_{max}$, together with the optimal $\frac{\chi_i^c}{\chi_a^c}$ ratio, allow estimating GPP via the familiar FvCB photosynthesis model (Farquhar et al., 1980) and applying Fick's law in turn yields $g_s^c$ and thus $g_s^s$. Finally, $g_i^s$ is obtained by scaling it to $V_{Cmax}$, as first
proposed by Berry et al. (2013):

$$g_i^s = 0.0012 V_{Cmax} .\tag{8}$$

Together, Eqs. (4-8) provide all the inputs for calculating the *LRU* via Eq. (3).

## 2.2 Data

The P-model has five environmental inputs - temperature, VPD, PAR, $C_a^c$, and air pressure - and was applied on the basis of
measured inputs determined within the frame of *in situ* leaf chamber measurements for validation, as well as on a global scale using fields of gridded inputs.

For validation we retrieved the datasets underlying the work by Kooijmans et al. (2019; only data from chamber #1 were used) and Sun et al. (2018) from the associated data repositories. Kooijmans et al. (2019) investigated the leaf-scale COS exchange of Scots pine (*Pinus sylvestris*) at the study site Hyytiälä in Finland (61°51′ N, 24°17′ E), while Sun et al. (2018) studied the
leaf-scale COS exchange of broadleaf cattail (*Typha latifolia*) at the San Joaquin freshwater marsh site in California/USA (33° 39′ N, 117° 51′ W). The major environmental difference between both studies is air temperature, which was ca. 7°C and 22°C for Kooijmans et al. (2019) and Sun et al. (2018), respectively.

The optimality implied in the P-model is likely to operate on multi-day to weekly time scales, as plants acclimate to the prevailing environmental conditions. Mengoli et al. (2022) devised an approach in which optimal (acclimated) values of $\xi$,
$V_{Cmax}$ and $J_{max}$ are calculated as running averages over the midday hours of the preceding 15 days, which are then used to estimate short-term (instantaneous) values of GPP, $g_s^c$, and the $\frac{\chi_i^c}{\chi_a^c}$ ratio. This approach was also applied to the leaf chamber data of Kooijmans et al. (2019) and Sun et al. (2018). The latter dataset did not include air pressure, which was inferred from elevation using the equation implemented in the P-model (Wang et al., 2017a). As *LRU* increases at low PAR values, while our interest is the light-saturated LRU, data were filtered for PAR > 1000 µmol m$^{-2}$ s$^{-1}$.

For application at the global scale, we calculated monthly climatologies of all inputs for the period 2001-2010 at a 0.05° resolution. Air temperature and VPD were taken from the Chelsea repository (version 2.1; Karger et al., 2018; Karger et al., 2017), incident PAR from Ryu et al. (2018), ambient $CO_2$ mole fractions from Cheng et al. (2022) and pressure was derived from a global digital elevation model included in the Chelsea repository using the equation implemented in the P-model (Wang et al., 2017a).

Usually, incident PAR in the P-model is multiplied with the (satellite-derived) fraction of absorbed PAR (fAPAR) as a simple means of leaf-to-canopy scaling (Stocker et al., 2020). Here, in order to compare to the available *LRU* studies, the interest is in the leaf-scale and the P-model was thus driven by incident PAR (leaf absorptance of PAR is included in the value of $\varphi_o$).



In contrast to the validation exercise described above, the monthly climatological inputs were used both for the acclimated and "instantaneous" calculations.


## 3 Results and Discussion

As shown in Fig. 1, Eq. (3) fed with inputs from the P-model overestimates the LRU of *Typha latifolia* by 26 %, while the LRU of *Pinus sylvestris* is underestimated by 33 %. The model also underestimates, by 57 % and 69 % respectively, the variability in measured LRU. While the P-model, or its predecessors, have been successfully validated in terms of the $\frac{c_i^c}{c_a^c}$ ratio

and $V_{Cmax}$ (Smith et al., 2019; Wang et al., 2017b), validation of LRU thus remains inconclusive and points to the urgent need for more *in situ* leaf gas exchange measurements from the major biomes of the world in order to truly understand to what degree the model is capable of reproducing the global patterns of LRU.

In order to exemplify the application of the model at global scale, Fig. 2 shows a global map of the growing season average

LRU. Simulated LRUs reach low values around 0.5 in the higher latitudes and, with a longitudinal mean of 1.34, peak in the tropics. The global median LRU amounts to 0.79 (95 % range: 0.53-1.41) and is thus roughly half of the value of 1.6 reported in earlier studies (Stimler et al., 2011; Stimler et al., 2010; Berkelhammer et al., 2014). As shown in Fig. 3, our values are also lower (by up to 34 %) than those reported recently by Maignan et al. (2021), who used the output of the process-based ORCHIDEE model to back-calculate global LRUs. Interestingly, these two completely independent estimates are highly

correlated across plant functional types ($R^2 > 0.93$), suggesting that both approaches reproduce similar patterns across the global bioclimatic space. It remains to be seen whether our, compared to previous estimates, low LRU values are able to resolve the longstanding conundrum in the global atmospheric COS budget, which is that estimates of a large land COS sink require an upward-tweak of the ocean source for the budget to close (Whelan et al., 2018). The magnitude of the required increases in the ocean source are however at odds with bottom-up estimates (Lennartz et al., 2017; Lennartz et al., 2021).


A sensitivity analysis, shown in Fig. 4, suggests that the simulated growing season averaged LRU is most sensitive to uncertainty in the $\frac{c_i^c}{c_a^c}$ ratio, especially in the tropics, where a 10 % increase may cause up to 40 % increase in LRU. In contrast, a 10 % increase in $g_s^s$ or $g_i^s$ results in maximum LRU changes of ± 8 %, the largest effect being observed in the higher latitudes. Here it is important to emphasize that the actual uncertainties in the $\frac{c_i^c}{c_a^c}$ ratio and $g_s^s$, which have been quantified for

decades from carbon isotope discrimination in plant biomass (Lloyd and Farquhar, 1994) and/or leaf gas exchange measurements, are likely to be much lower than the ones of $g_i^s$. For the latter much less data is available and its parameterization (Eq. 8) is poorly constrained by it (Berry et al., 2013). Using the process-based model SiB4, Kooijmans et



 

al. (2021) inferred the coefficient which scales $g_i^s$ to $V_{Cmax}$ to vary between ca. 0.0005 and 0.003 for selected field study sites, which is 60 % lower and 150 % higher compared to the reference value of 0.0012 put forward by Berry et al. (2013).


## 4 Conclusions

Accurate knowledge of the LRU is prerequisite to using Eq. (2) for estimating GPP (Wohlfahrt et al., 2012). While earlier work suggested the LRU, under saturating light conditions, to be confined to ca. 1.6 (Stimler et al., 2011; Stimler et al., 2010; Berkelhammer et al., 2014), current cumulative evidence suggests the LRU to be more variable (Whelan et al., 2018) and Sun

et al. (2022) recently concluded from a theoretical analysis "there is no guarantee for LRU to converge to a narrow range across species". Inspection of Eq. (3) shows that convergence to a universal value would require the $\frac{c_i^c}{c_a^c}$ and $\frac{g_s^s}{g_i^s}$ ratios to be constant or compensating changes between the two. At present we have no evidence to support either of these scenarios. Rather, our global simulations, based on plant optimality principles, suggest the LRU to predictably vary (Fig. 2), reflecting spatial patterns in the $\frac{c_i^c}{c_a^c}$ ratio, $g_s^s$ and $g_i^s$ (Fig. 4). We recognize that our values, in the range between 0.5 and 1.4, are low

compared to those used in previous global assessments. We thus advocate, until more empirical measurements become available for validating our simulations, forward and inverse modellers to adopt our values/approach in order to examine whether these help to reconcile some of the long-standing inconsistencies in the global COS budget.





*Code and data availability:*

The datasets of Kooijmans et al. (2019) and Sun et al. (2018) are available from https://zenodo.org/record/1211481#.Y0VnjC-2276 and https://datadryad.org/stash/dataset/doi:10.15146/R37T00, respectively. Air temperature, VPD and the digital elevation model were taken from https://chelsa-climate.org, incident PAR from https://www.environment.snu.ac.kr/bess-rad, $CO_2$ mole fractions from https://zenodo.org/record/5021361#.Y0Vmz0zP2gM. All data and the Matlab scripts used for processing these and creating the figures can be found at https://zenodo.org/record/7185592#.Y0a1vEzP2Ht. For easy adoption

of our global LRUs by modellers we provide these as monthly climatological means, averaged for the period 2001-2010, at 0.05° resolution (lru_pmodel_global_monthly_climatology.nc).

*Author contributions:*

GW conceived the study and conducted the simulations and analyses with the help of AH. All authors contributed to the writing

of the manuscript.

*Competing interests:*

The authors declare no competing interests.

*Acknowledgements:*

The authors thank Colin Prentice and Julia Mengoli for patiently answering questions regarding the P-model. Fabienne Maignan and Camille Abadie are acknowledged for providing the ORCHIDEE plant functional type map.

*Financial support:*

This study was financially supported by the Austrian Science Fund (FWF) through grant P35737.



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





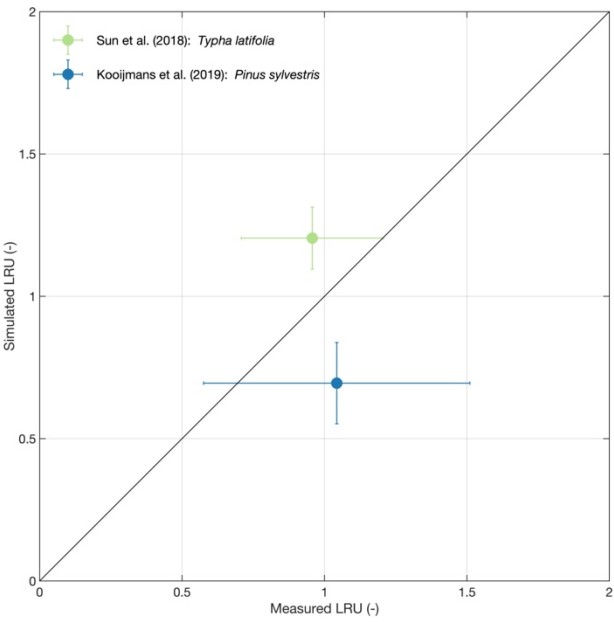

**Figure 1: Model validation. Symbols and error bars represent means and their standard deviations, respectively.**





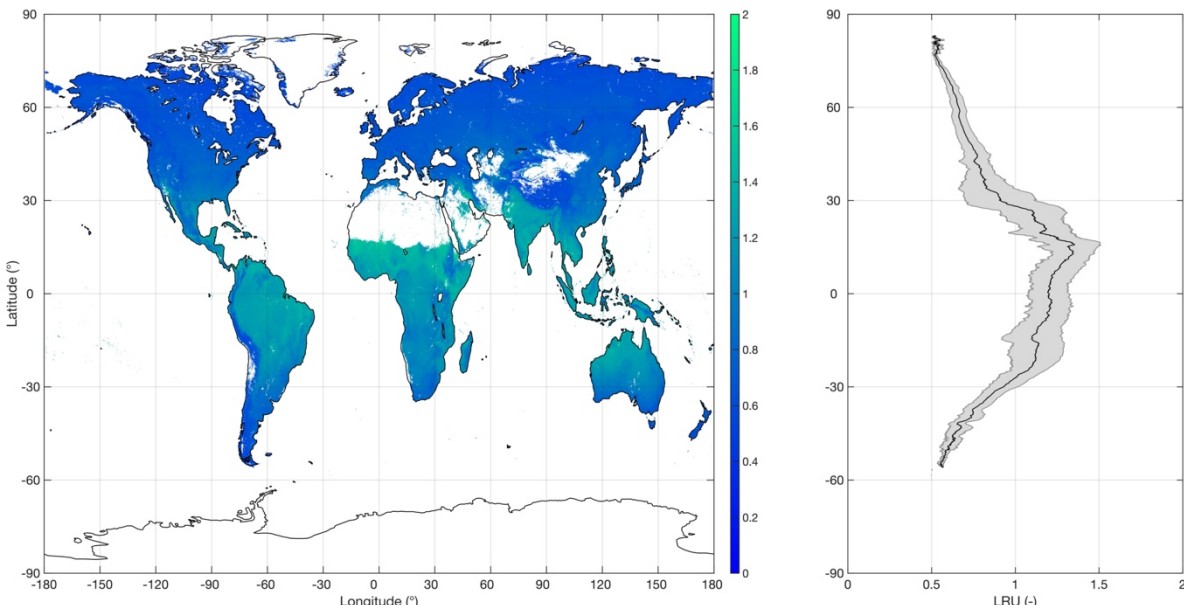

**Figure 2: Simulated growing season (monthly average air temperature above 0°C) average global LRU (left) and longitudinal averages (right). The solid line and the shaded area in the left plot represent means and their standard deviations, respectively.**



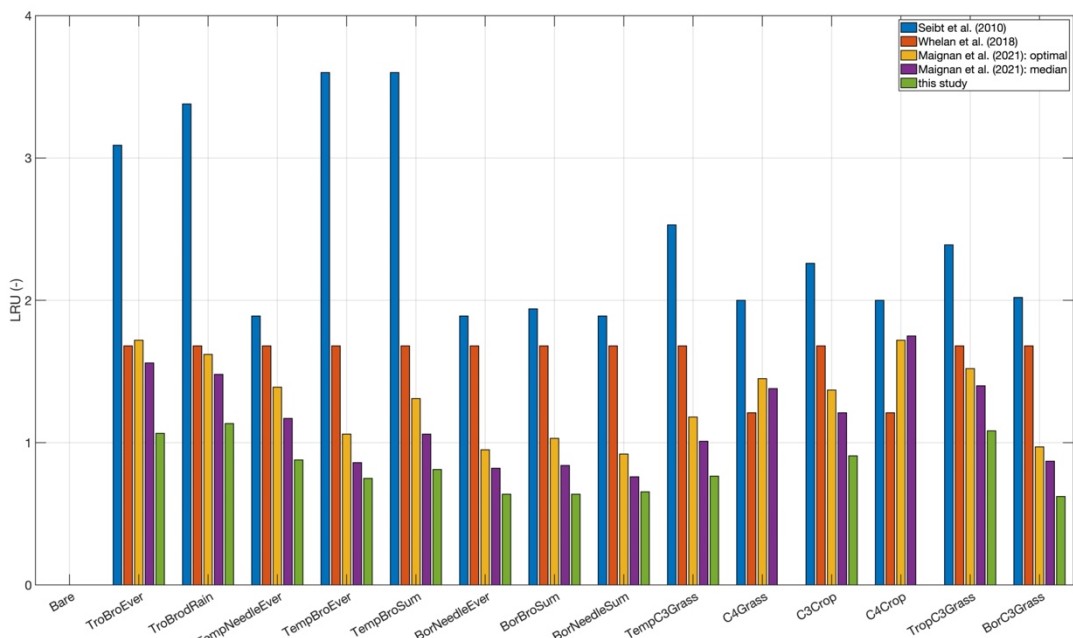

**Figure 3: Comparison to published global LRU values averaged by plant functional type. Published LRU values were taken from Table 1 in Maignan et al. (2021). The plant functional type classification corresponds to the one used in the ORCHIDEE model (Krinner et al., 2005). No values are given for the C₄ grass and crop plant functional types, since the model is applicable to C₃ species only.**





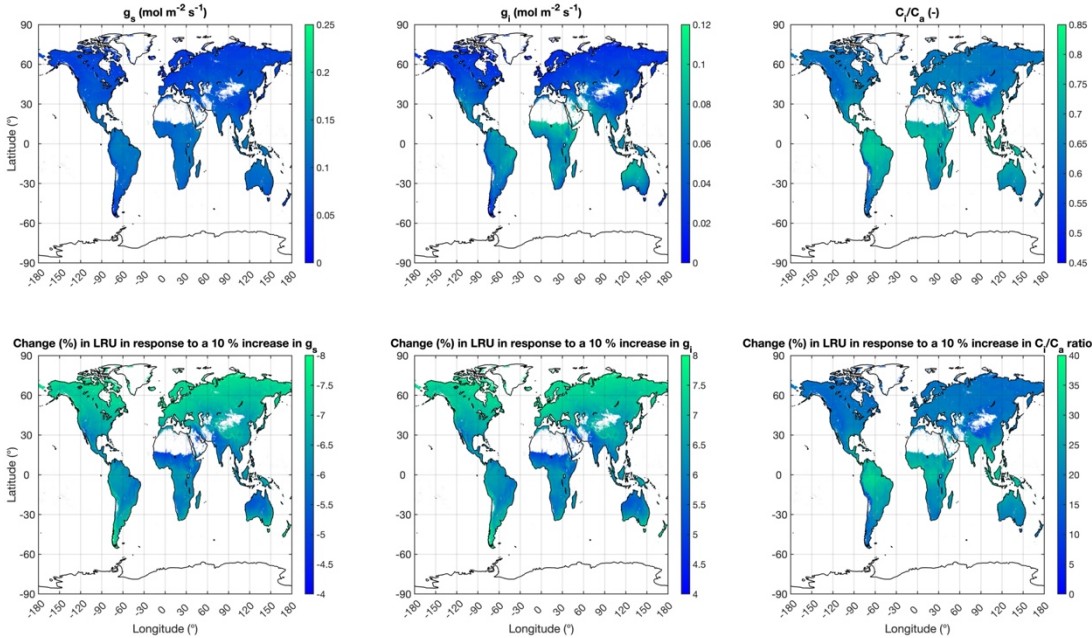

295

**Figure 4: Simulated growing season average stomatal conductance to COS ($g_s$; upper left panel), internal conductance to COS ($g_i$; upper middle panel) and the internal to ambient $CO_2$ mole fraction ratio ($C_i/C_a$; upper right panel), as well sensitivity of LRU to a 10 % increase in stomatal conductance to COS (lower left panel), internal conductance to COS (lower middle panel) and the internal-to-ambient $CO_2$ mole fraction ratio (lower right panel).**