# Peer review of "Technical note: Novel estimates of the leaf relative uptake rate of carbonyl sulfide from optimality theory"

_Biogeosciences, 2022_

## Author Response (AR1)

**Summary**

Reflecting the reviewer and editor comments, the major changes in the manuscript are:

1. Both reviewers commented on the sensitivity analysis. Reviewer #1 suggested to study changes in the major parameters instead of the processes as we had done initially, while reviewer #2 suggested to use a more realistic uncertainty range for $g_i$. We have followed both reviewer comments and now show the uncertainty to the major calibrated model parameters, including a realistic uncertainty range for $g_i$ (or rather the alpha parameter which governs $g_i$). The global maps of $g_s$, $g_i$ and $C_i/C_a$, which were included with the sensitivity analysis in former Figure 4, have been moved to the supplement.
2. Following the comment by reviewer #2 about new evidence regarding $g_i$, we have changed the alpha parameter from 1200e-6 to the mean value found by Kooijmans et al. (2021), 1616e-6. As a consequence, all LRU values increased slightly and the corresponding figures and all text have been updated accordingly.
3. The methods section has been partially rewritten and reorganized in order to more clearly present our approach.

**Editor comments to the authors:**

I would like the thank firstly the reviewers for their high quality reviews, and secondly yourselves (the authors) for your detailed response and patience through the holiday season.

It is my recommendation that you proceed with resubmission of the updated manuscript that follows the additions and revisions that you outlined in your responses to the reviewers.

In addition, I would like to add the following recommendations:

Re: Response to Reviewer 1: L80-81.

This is an effective explanation of the links (or lack thereof) between LRU and C*. I would recommend adding a statement along these lines to the manuscript or perhaps better would be in an SI so as to not disrupt the flow of the manuscript. There may be some additional elaborative information from additions/adjustments relating to other comments that might also fit into a supplemental information. However, if the authors believe the manuscript is better served without an SI, I am happy to accommodate this with an explanation.

*R: an explanation why LRU is not sensitive to c\* has been added to the main text of the manuscript*

Re: Response to Reviewer 2: Electron Transport Limitation.

While this seems like a reasonable explanation, I would like to give the opportunity to the reviewers to comment again after resubmission regarding this point.

*R: fine*

Re: Response to Reviewer 2: Internal conductance of OCS.

Please ensure the discussion is also updated to account for the expanded sensitivity analysis.

*R: yes, the discussion has been updated accordingly*

Re: Response to Reviewer 2: Midday hours photosynthesis.

I am in agreement that expanding on photosynthetic reduction due to stresses of high heat, vapor pressure and peak solar radiation intensity likely goes beyond the scope of the paper. I would suggest two options: (1) if it is not too challenging, add a simulation that changes midday photosynthesis to peak photosynthesis and comment on the differences (some of this may serve in an SI), or (2) in the least add a statement on this potential limitation / inclusion of bias to the manuscript.

*R: we have followed option (2) and discuss the limitations of this approach*

**Reviewer #1**

This technical note proposes a new theoretical approach to provide estimates of the Leaf Relative Uptake (LRU) of carbonyl sulfide (COS) with respect to $CO_2$, along large-scale bioclimatic gradients. It is based on plant optimality and coordination hypotheses. The LRU is useful to estimate biosphere COS fluxes based on gross primary productivity (GPP) and is often used for atmospheric inversions against COS atmospheric concentrations. The paper is well built and well written, with a literature review quite up to date, and clearly defined objectives. Plus, the derived LRU maps, as well as the scripts, are made available on a repository, which is quite commendable.

This study will be of interest to the whole COS community. The new estimates are intriguingly quite low as compared to previous ones, this will most certainly fuel interesting discussions to understand why, and what the consequences are for the biosphere COS and GPP budgets, for the closure of the global COS budget and for atmospheric inversions. As often in the COS field, the authors advocate for more in situ observations in more biomes, needed to correctly evaluate the predictions of this new framework. I recommend the publication of this study and have only minor comments.

L63: The P-model is applicable only to C3 plant species. The authors should add something in the legend of Figure 2 or mask grid cells where C4 plants are predominant.

*R: Figure 2 has been modified to indicate the presence of C4 plant species by means of the transparency level of the color coding*

L68: The authors could add a short analysis to quantify the sensitivity of LRU to the beta parameter. Wang et al. (2017) indeed show that beta (with a former slightly different formulation) varies when they account for the mesophyll conductance, and they also suggest that beta is assumed a constant but could be varying with plant functional traits.

*R: Figure 4 has been modified and now shows a sensitivity analysis of LRU with regard to the alpha and beta parameter; see below for why the sensitivity to the c\* parameter was not included*

L80-81: c\*=0.41 seems to be based on two numbers (Jmax/Vcmax = 1.88 and chi = 0.8) following Stocker et al. (2020). Stocker et al. (2020) also mentions that Smith et al. (2019) use another Jmax modelling. Again, as stated by Wang et al. (2017), c\* could vary with functional traits and the authors could add a sensitivity analysis of LRU to c\*.

*R: the c\* parameter affects both Vcmax and Jmax (see Eqs. 6 and 7 in the manuscript); a reduction in c\* increases Vcmax and thus (remember that the model assumes co-limitation by RUBISCO and electron transport on longer time scales) GPP and thus, because Ci is unaffected by c\* (Eqs. 4 and 5), causes a proportional increase in gs; since gi is proportional to Vcmax via Eq. 8, the ratio of gs/gi in Eq. (3) remains unaffected by changes in c\* - in other words: LRU is not sensitive to changes in c\* and was thus omitted from the sensitivity analysis; a corresponding explanation was added to the manuscript*

L92: "Kooijmans et al. (2019; only data from chamber #1 were used)": is there a specific reason why the data from chamber #2 were discarded from the validation?

*R: no specific reason … Figure 1 and the corresponding text were updated to include both chambers of the Kooijmans et al. (2019) study*

L104: "data were filtered for PAR > 1000 µmol m-2 s-1". Could this (partly) explain why the authors find lower LRU values, as compared to estimates by land surface models that calculate mean LRU values over all PAR conditions? Could the authors quantify the effect of this filtering?

*R: yes, the filtering for light-saturated conditions may explain the higher values of Seibt et al. (2010) and Whelan et al. (2018), as the data underlying these studies were not filtered for radiation; with regard to the comparison to the results by Maignan et al. (2021), the bigger issues is a difference in scale, as their values are integrated over the depth of the plant canopy and thus, because of the decrease in radiation and VPD, should be higher; the text was updated to include a discussion of these issues*

L110-112: This part is not crystal-clear, and neither is the corresponding argumentation in Stoker et al. (2020). Yet I believe it is fundamental to explain what is leaf-level and what is canopy-level, and what information is exactly put in the $\delta \cdot ce_0$ in this study (as opposed to the P-model version). Plus, later the authors indeed compare their results both with leaf-level observations and with canopy-level LRU estimates from land surface models. The authors should detail and clarify this section, and maybe say something on how a canopy-level LRU compares with a leaf-level LRU, is one systematically higher than the other?

*R: the entire methods section was re-organized and partially re-written in order to convey more clarity; it should now be clear that model simulations represent the leaf-scale only and in the one case when we compare against the canopy-integrated LRU simulations by Maignan et al. (2021) we discuss the issue of leaf-to-canopy scaling and why canopy-scale LRUs must be expected to be higher than at leaf scale*

L129-131: The authors mention a large correlation across plant functional types between the LRU of this study and the ones derived from the ORCHIDEE land surface model. I guess this is expected as both approaches are using very similar models driven by meteorological fields. The figure seems to show that the difference is not constant but proportional to LRU. A scatter plot with a regression line could help in this analysis.

*R: as suggested, a scatter plot was added to Figure 3, which shows a positive relationship between bias and LRU – this finding is discussed in the updated text; while both models are driven by presumably similar meteorological fields, the stomatal conductance model used in ORCHIDEE, a variant of the Ball-Berry-Woodrow model, is quite different and we will modify the text in order to emphasize this point*

Typos

L63-64: hypotheses (plural, twice)

*R: corrected*

L83, L101: The letter chi is (erroneously?) used instead of c for the ratio of concentrations.

*R: corrected*

**Reviewer #2**

The study by Wohlfahrt et al. (2022) proposes a new way to constrain the variability of OCS leaf relative uptake (LRU) ratio across plant functional types (PFTs) and climatic gradients by fusing an LRU model to the eco-evolutionary optimality framework (Maire et al., 2012, PLoS ONE; Prentice et al., 2014, Ecol. Lett.). LRU is a key parameter to translate leaf OCS uptake into constraints on gross photosynthesis. However, there are limited observations to inform LRU variability with climate and across species. This study leverages the optimality theory to bypass the data gap, allowing LRU to be predicted in hitherto unobserved biomes. If the prediction holds against future observations, the results can help calibrate land surface models and provide the LRU input for atmospheric inverse modeling of regional and global OCS fluxes. The study will be of interest to the OCS community as well as the broader photosynthesis research community.

While I do not question the validity of the main conclusions, there seem to be a few assumptions involved in deriving the "optimal" LRU, which need to be articulated and examined. A few technical issues also need to be addressed to make the results more robust. Given that the other reviewer has commented extensively on the P-model, here, I focus on other aspects.

1. Electron transport limitation: The Prentice et al. (2014) optimality model assumes photosynthesis is Rubisco-limited. This assumption may not hold for shoulder seasons and high-latitude sites (e.g., boreal forests) in which photosynthesis is often light (electron transport) limited. The latter case, as pointed out by Prentice et al. (2014), may be examined by substituting $\xi$ with the electron transport-limited value given by Medlyn et al. (2011) Global Change Biol.

*R: while the reviewer is correct in that the Prentice et al. (2014) model assumes photosynthesis to be Rubisco-limited, the P-model in the version by Mengoli et al. (2022), which is used here, does not so as it additionally adopts the co-ordination theory, which results in co-limitation by both Rubisco and electron transport over the (longer) time scale on which plants acclimate to the prevailing environmental conditions; on short (instantaneous) time scales, photosynthesis however is either limited by electron transport or Rubisco as is the original FvCB model.*

2. The model validation shown in Fig. 1 is too broad-brush to be useful. Looking at this figure, we can tell the direction of the mean bias, but we have no idea how well the simulated LRU values capture the variability in the observed values. I recommend showing a scatter plot and reporting the mean bias, RMSE, and $R^2$ for each data set.

*R: Figure 1 was updated according to the reviewer comment by including the underlying raw data – following a comment by reviewer #1, data of chamber #2 from Kooijmans et al. (2019) were additionally added; the requested statistical metrics were added to the text*

3. The internal conductance of OCS ($g_i$), which includes components of mesophyll conductance and carbonic anhydrase activity, is not constrained by the optimality framework. Thus, $g_i$ may contribute greatly to the uncertainty in LRU. Although the authors attempted a sensitivity test by varying this parameter by 10%, it is not enough, given that Kooijmans et al. (2021) find that optimized $g_i/V_{c,max}$ ratios can deviate a lot from the original Berry et al. (2013) parameterization. My suggestion to mitigate the problem of $g_i$ uncertainty would be to test the sensitivity of LRU to $g_i$ over a wider range of $g_i/V_{c,max}$ ratio, from 600 to 3000, encompassing the range shown in Fig. 4 of Kooijmans et al. (2021).

*R: the sensitivity analysis regarding the gi parameter was improved by using the standard deviation of the alpha value (±562) given in Kooijmans et al. (2021); in doing so we have also replaced the alpha proposed by Berry et al. (2013) (1200) with the mean value by Kooijmans et al. (2021) (1616)*

Minor comments

- L11: I would leave out "alternative" because readers may not have known other tracers of GPP.
*R: changed as suggested*

- L12: "LRU" -> "light-saturated LRU" - Given that the prediction focuses on light-saturated LRU, I would make the distinction early on so that readers know what they should be comparing the LRU values to.
*R: changed as suggested*

- L14: "0.5–1.4" - What is the statistical distribution of LRU values across all grid cells? How does it compare with Fig. 2 in Whelan et al. (2018) Biogeosci.?
*R: to address this question we have added the data from Figure 2 in Whelan et al. (2018) to our Figure 2*

- L56: "the lack of a suitable theoretical framework" + "to predict LRU a priori"
*R: changed as suggested*

- L96: Are the temperatures reported here mean annual temperatures or averaged over the campaign periods?
*R: these are averages of the measurement campaigns –added corresponding clarification to the text*

- L100: Using midday hours to determine the optimal values may create a bias, because photosynthesis is often suppressed around midday due to stressed conditions under high light or high vapor pressure deficit. Why not use data at the hour of peak photosynthesis (whenever it is) to determine these parameters?
*R: this is a very interesting comment as it addresses the issue of the time scale over which plants would acclimate to the prevailing environmental conditions, which is not a priori apparent; since this paper is about the application of the P-model for estimating LRU rather than improving the P-model, we believe that changing the model is out of scope, but we discuss this potential bias in the revised text*

- Fig. 2: Not the best color scheme because it does not have a strong contrast between the minimum and maximum values. On the right panel, consider adding the observed values from Sun et al. (2018) and Kooijmans et al. (2019) for visual comparison.
*R: the colormap was changed to one (Batlow) in the package recommended by the journal's guide for authors; following a comment by reviewer #2 we have added the LRU distribution from Whelan et al. (2018) to Figure 2, which is better suited to put our results into perspective*

- Fig. 3 does not seem to compare apples to apples. Seibt et al. (2010) did not limit LRU to light-saturated values, hence showing higher values. LRU values in Maignan et al. (2021) are modeled, and should not be treated as observations. These caveats should be noted in the figure caption.
*R: correct – because Seibt et al. (2010) and Whelan et al. (2010) did not explicitly filter for PAR, their values must be expected to be higher – this was included in the discussion; the fact that LRU values from Maignan et al. (2021) are modelled and represent canopy-integrated values is now also discussed in the revised text*

- Fig. 4: Same as Fig. 2, the color scheme lacks contrast.
*R: see reply to comment above*

---

## Author Response (AR2)

**Associate Editor comments:**

Thank you kindly for your excellent responses and patience. I was able to confirm that reviewer 1 has approved your changes. The reviewer asked to make one minor comment: "In the ORCHIDEE-COS version, the decrease of light availability is represented only within the canopy. It does not compute the variables that would be required to represent a vertically varying VPD." Please review this with reference to lines L153-155 in the updated manuscript and advise if you will make any minor adjustment. Otherwise, I am happy to inform you the manuscript will be accepted for publication.

**Author response:**

We thank reviewer #1 and the associate editor for the swift turnaround time. We thank reviewer #1 for clarifying that the ORCHIDEE model does not represent the vertical within-canopy VPD profile and that our statement was thus incorrect. We have reformulated l. 153-156 to: "In this comparison it should be noted that ORCHIDEE integrates LRU over the depth of the plant canopy. PAR availability, together with VPD the major driver of short-term variability in LRU, typically decrease with canopy depth and since LRU is negatively related to PAR (e.g. Kooijmans et al., 2019; Kohonen et al., 2022), canopy-integrated LRU is expected to be larger than leaf-scale LRU at the top of the canopy (Sun et al., 2022)."

No other changes have been made to the manuscript.

On behalf of all co-authors,

Georg Wohlfahrt